# Intravesical CD74 and CXCR4, macrophage migration inhibitory factor (MIF) receptors, mediate bladder pain

Shaojing Ye[1ᴼ], Fei Ma[1ᴼ], Dlovan F. D. Mahmood[1], Katherine L. Meyer-Siegler[2], Raymond E. Menard[2], David E. Hunt[1], Lin Leng[3], Richard Bucala[3], Pedro L. Vera[1,4]*

1 Research & Development, Lexington VA Health Care System, Lexington, KY, United States of America,
2 Department of Natural Sciences, St Petersburg College, St Petersburg, FL, United States of America,
3 Department of Internal Medicine, Yale University, New Haven, CT, United States of America, 4 Department of Physiology, University of Kentucky, Lexington, KY, United States of America

ᴼ These authors contributed equally to this work.
* Pedro.Vera@va.gov

**Data Availability Statement:** All relevant data are within the manuscript and its Supporting information files.

## Abstract

### Background

Activation of intravesical protease activated receptor 4 (PAR4) leads to release of urothelial macrophage migration inhibitory factor (MIF). MIF then binds to urothelial MIF receptors to release urothelial high mobility group box-1 (HMGB1) and elicit bladder hyperalgesia. Since MIF binds to multiple receptors, we investigated the contribution of individual urothelial MIF receptors to PAR4-induced HMGB1 release *in vivo* and *in vitro* and bladder pain *in vivo*.

### Methodology/Principal findings

We tested the effect of intravesical pre-treatment with individual MIF or MIF receptor (CD74, CXCR4, CXCR2) antagonists on PAR4-induced HMGB1 release *in vivo* (female C57/BL6 mice) and *in vitro* (primary human urothelial cells) and on PAR4-induced bladder hyperalgesia *in vivo* (mice). In mice, PAR4 induced HMGB1 release and bladder hyperalgesia through activation of intravesical MIF receptors, CD74 and CXCR4. CXCR2 was not involved in these effects. In primary urothelial cells, PAR4-induced HMGB1 release through activation of CD74 receptors. Micturition parameters in mice were not changed by any of the treatments.

### Conclusions/Significance

Urothelial MIF receptors CD74 and CXCR4 mediate bladder pain through release of urothelial HMGB1. This mechanism may set up persistent pain loops in the bladder and warrants further investigation. Urothelial CD74 and CXCR4 may provide novel targets for interrupting bladder pain.

**Funding:** PLV (DK121695) RB (AR049610) National Institutes of Health The funders had no role in study design, data collection and analysis, decision to publish, or preparation of the manuscript.

**Competing interests:** The authors have declared that no competing interests exist.

## Introduction

Urothelial cells constitutively express all protease activated receptors (PAR), PAR1, PAR2, PAR3 and PAR4 [1]. Moreover, urothelial cells also constitutively express macrophage migration inhibitory factor (MIF) [2]. MIF is proinflammatory cytokine involved in several inflammatory and autoimmune disease states (e.g. rheumatoid arthritis [3]) and also more recently implicated in mediating pain [4, 5].

We previously showed that intravesical activation of PAR4 receptors resulted in acute increased abdominal mechanosensitivity (bladder hyperalgesia; BHA) at 24 hr post intravesical PAR4 activation with no changes in awake micturition parameters or major histological evidence of bladder inflammation. We also showed that PAR4-induced BHA was mediated by MIF since urothelial MIF was released by PAR4 treatment and PAR4-induced BHA could be blocked by systemic treatment with a MIF antagonist or was absent in MIF deficient mice [6, 7].

Finally, we reported that intravesical activation of PAR4 receptors induced release of urothelial HMGB1 that resulted in bladder pain by acting on intravesical TLR4 receptors [8]. Therefore, in this model of acute BHA, activation of urothelial PAR4 receptors triggers release of urothelial MIF, presumably to bind to urothelial MIF receptors and induce release of urothelial HMGB1. Urothelial HMGB1 then produces bladder pain through activation of bladder TLR4 receptors [8].

MIF exerts its actions through binding a cell-surface multireceptor complex that includes CD74 (canonical receptor) and also CXCR4 and CXCR2 [9]. CXCR4 and CXCR2 are not exclusive receptors for MIF and they also bind CXCL12 and CXCL8, respectively [10]. We and others showed that all of these receptors are present in the urothelium [11–15]. However, whether MIF receptors at the bladder directly mediate PAR4-induced BHA or the contribution of specific MIF receptors to PAR4-induced BHA or to urothelial HMGB1 release remain to be determined.

Therefore in the current study we tested the effects of intravesical pretreatment with antagonists to MIF or individual MIF receptors on PAR4-induced BHA and on PAR4-induced HMGB1 release in our animal model of BHA. Moreover, we characterized the contribution of individual MIF receptors in mediating PAR4-induced MIF and HMGB1 release from primary human urothelial cells *in vitro*. In addition, we further characterized our model of PAR4-induced acute BHA by examining whether intravesical PAR4 alters micturition by studying cystometric changes (in anesthetized mice) and whether intravesical PAR4 treatment produces changes in bladder levels and/or expression of inflammatory cytokine (TNF-$\alpha$, IL-1$\beta$) that may be indicative of bladder inflammation.

## Materials and methods

All animal experiments were approved by the Lexington VA Health Care System Institutional Animal Care and Use Committee (VER-19–005-AF) and performed according to the guidelines of the National Institutes of Health. C57/BL6 female mice were purchased from Jackson Laboratory (Jackson Laboratory, Bar Harbor, ME) and were 12–14 weeks of age at the time of the experiments. Mice were housed in standard rodent cages in rooms with a 14 hr on (7 am); 10 hr off (9 pm) light cycle with *ad libitum* access to food (Envigo Teklard Global 2018) and water.

### Acute bladder hyperalgesia (BHA) model and test of antagonist pretreatment

We used intravesical administration of PAR4-activating peptide (PAR4-AP) to induce acute bladder hyperalgesia (BHA) as described earlier [6, 7, 16]. In addition, we tested whether intravesical pre-treatment with different antagonists block PAR4-induced BHA (Fig 1).

- Day 0 (Baseline):

VF → isoflurane ——————————————→ Recover

*Pre-Treat 10 min* ——————→ **Treat 1hr**

PBS/Vehicle
Anti-MIF
Anti-CD74 ——————→ **PAR4-AP or Scramb**
CXCR4 Ant
CXCR2 Ant

- Day 1 (24 hr post):

VF ——————→ VSOP (Awake) ——————→ Isoflurane
**or**                           collect bladder
CMG (Anesthetized)

**Fig 1. Experimental protocol for inducing and antagonizing PAR4 bladder hyperalgesia (BHA).** Mice were administered a baseline abdominal von Frey (VF) test and then anesthetized with isoflurane. Bladder was emptied through a transurethral catheter and MIF or MIF-receptor antagonists were instilled for 10 min prior to instillation of PAR4 activating peptide (PAR4-AP; to elicit BHA) or a scrambled peptide (Scramb; as control). After 1 hour the intravesical fluid was collected and mice were allowed to recover. Next day, mice were again tested for VF and micturition volume and frequency were recorded in awake mice using the Voided Stain on Paper Method (VSOP). Alternatively, mice were anesthetized and instrumented for cystometry (as described). At the end of the study, bladders were collected for histology and further analysis.

Briefly, mice were anesthetized with isoflurane and transurethrally catheterized (PE10, 11 mm length). Urine was drained by gently applying pressure to the lower abdomen. Bladders were slowly instilled with 50 $\mu$l each as pretreatment (Fig 1): MIF antagonist (15 $\mu$g anti-MIF monoclonal antibody, or its isotype control mIgG1); or CD74 antagonist (15 $\mu$g anti-mouse CD74 monoclonal antibody, or its isotype control rIgG2b); MIF098 as a MIF/CD74 binding inhibitor (200 $\mu$g, or 40% (2-hydroxypropyl)-P-cyclodextrin and 10% PEG400 as vehicle); CXCR4 antagonist (50 $\mu$M AMD3100 or a vehicle PBS control); or CXCR2 antagonist (6 $\mu$g SB225002; Tocris, or a vehicle 0.1% DMSO in PBS control) for 10 minutes before the instillation of either PAR4-AP (AYPGKF-NH2) or a corresponding scrambled peptide (as a PAR4 control; YAPGKF-NH2). The peptides (100 $\mu$M) were dissolved in sterile phosphate-buffered saline (PBS, pH 7.4, 100 $\mu$l) and remained in the bladder for 1 hour. The intravesical fluid was then collected, treated with phosphatase and protease inhibitors and stored at -80˚C prior to analysis. Group size (N = 6/treatment) was calculated by sample-power analysis using R [17] (mean sample difference = 50%, s.d. = 25% of mean, two-sided t-test; estimated power = 87%).

## Abdominal mechanical sensitivity

We tested abdominal mechanical hypersensitivity in mice as previously described [18–20]. Briefly, 50% mechanical threshold [21–23] was calculated by measuring the response to von Frey fibers (0.008, 0.02, 0.07, 0.16, 0.4, 1.0, 2.0 and 6.0 g) applied to the lower abdominal region. A positive response (an indicator of bladder hyperalgesia) was defined as any one of three behaviors: (1) licking the abdomen, (2) flinching/jumping, or (3) abdomen withdrawal. Whenever a positive response to a stimulus occurred, the next smaller von Frey filament was applied. Otherwise, the next higher filament was applied. 50% thresholds were measured at

baseline (prior to any treatment) and approximately 24 hours after bladder pre-treatments and treatments (Fig 1).

## Voided stain on paper (VSOP): Micturition parameters in awake mice

We measured micturition volume and frequency in awake mice using the VSOP method [24] as described earlier [6–8, 16]. Briefly, 24 hours after intravesical treatment and the abdominal mechanical behavioral test (Fig 1) mice were placed in a plastic enclosure individually with freedom to move around and access to water. Filter paper was placed under the cage to collect urine during a 3-hour observation period. Micturition volumes were determined by linear regression using a set of known volumes. Micturition frequency is reported as the number of micturitions per 3-hour observation period.

## Cystometry: Micturition parameters in anesthetized mice

Twenty-four hrs after intravesical treatment, mice were anesthetized (urethane; Sigma-Aldrich; 1.1–1.3 mg/kg in saline; i.p.). An abdominal incision exposed the bladder and a catheter (PE50) was inserted into the bladder dome and secured with a purse string suture. The catheter was connected to an infusion pump (WPI; Sarasota, FL; model sp100i) for sterile saline infusion while connected to a pressure transducer. Bladder pressure was recorded using Spike2 (CED; Cambridge, England).

One hour after catheter insertion, bladders were emptied and filled with room temperature sterile saline at the rate of 0.8 ml/hr. Single cystometrograms were conducted three times with bladder emptying in between cystometrograms. The following parameters were measured: Volume infused, micturition threshold, peak pressure and micturition volume. A piece of filter paper placed at the urinary meatus and weighed before and after micturition was used to record micturition volume. Voiding efficiency was calculated as the percentage of micturition volume/infused volume.

Following single cystometrograms, saline was continuously infused (0.8 ml/hr) for 30 min to elicit continuous contractions. Intercontraction intervals were calculated for each mouse. Group size (N = 8 in each condition) was calculated by sample-power analysis using R [17] (mean sample difference = 40%, s.d. = 25% of mean, two-sided t-test; estimated power = 95%).

## *in vitro* experiments: Primary human urothelial cells

Normal bladder epithelial cells (from the apex) were obtained from LifeLine Cell Technology (Frederick, MD; FC-0079; 24 yr old African American male; cause of death head trauma, GSW). Cells were plated in 24 well plates at a seeding density of $5x10^4$ cells/cm$^2$ for 72 hours (2 doublings) using UroLife basal medium supplemented as recommended by the manufacturer (LifeLine; #LL-0071). Medium was removed and exchanged with PBS containing 1% glutamine and a MIF antagonist (ISO-1; 100 $\mu$M) or specific MIF-receptor antagonist (MIF098 10 $\mu$M; AMD3100 5 $\mu$M) for 15 minutes. Followed by PAR4–AP or scramble peptide at 50 $\mu$M and incubation for an additional 60 minutes. Supernatants were collected and cell lysates produced by the addition of NP-40 buffer. All samples were treated with phosphatase and protease inhibitors and stored at -20˚C prior to analysis.

## Enzyme-linked immunosorbent assay (ELISA)

HMGB1 levels in intravesical fluid were assayed using a mouse HMGB1 ELISA kit (MBS776360), while HMGB1 levels in culture media were evaluated using a human HMGB1 ELISA kit (MBS701378) per the manufacturer's protocols (MyBioSource, San Diego; CA).

Bladder tissue was homogenized in the presence of protease inhibitors (Halt Protease and Phosphatase Inhibitor Single-Use Cocktail, Thermo Fisher Sci.). Bladder levels of interleukin IL-1$\beta$ (MBS776446) and tumor necrosis factor-$\alpha$ (MBS161138) were determined as per the manufacturer's protocols (MyBioSource) and normalized to lysate protein concentrations (as determined by BCA).

## Real-time PCR

Total RNA was extracted from mouse bladder tissue using AMBION Trizol and DNA was removed by DNase (AM1906, Invitrogen). All RNA samples were quantified with a Nano-DropTM 1000 spectrophotometer (Thermo Scientific). 1 $\mu$g of RNA was reverse transcribed to cDNA using a commercial reverse transcription system (A3500, Promega, Madison, WI). SYBR green (330509, Qiagen Scientific) was utilized with primers (Qiagen, Germantown, MD) for the target genes (TNF$\alpha$, PPM03113G; IL-1$\beta$, PPM03109F) and 5 housekeeping genes (Rn18s, PPM72041A; GAPDH, PPM02946E; Rpl32, PPM03300B; Hsp90ab1, PPM04803F; Actb, PPM02945B) to quantify the level of mRNA in bladder tissue from PBS-Scramble and PBS-PAR4 treated mice. Thermal cycling in a StepOnePlus Real-Time PCR System (Applied Biosystems; Grand Island, NY) proceeded as follows: 10 minutes at 95˚C, followed by 40 cycles of denaturation (15 seconds at 95˚C) and annealing / extension (60 seconds at 60˚C). Expression changes were calculated using the $\Delta\Delta$CT method. In order to increase the rigor and reproducibility of expression changes, we evaluated five common housekeeping genes for potential changes due to experimental treatment. Mean expression of the three housekeeping genes with robust expression and greatest expression stability between the groups (18S rRNA, Actb and Hsp90ab1) was used to normalize target gene expression [25, 26]. Data are from duplicate wells of each bladder sample.

## Histological measurements

At the end of study mice were anesthetized with 3–4% isoflurane, bladders were rapidly removed and segments were either placed in 4% buffered formaldehyde for histology or frozen for further analysis. Bladder paraffin sections (5 $\mu$m) were processed for routine hematoxylin and eosin (H&E) staining. H&E stained sections were evaluated by a pathologist blinded to the experimental treatment and scored separately for edema and inflammation according to the following scale: 0, No edema/no infiltrating cells; 1, Mild submucosal edema/few inflammatory cells; 2, Moderate edema/moderate number of inflammatory cells; 3, Frank edema, vascular congestion/many inflammatory cells, as per our previous studies [6–8, 16, 18].

## Reagents

PAR4-AP (AYPGKF-NH2)] and corresponding scrambled peptide (YAPGKF-NH2) as control were from Peptides International, Inc. (Louisville, KY). Anti-MIF monoclonal antibody, isotope control (mIgG1) and anti-CD74 monoclonal antibodies were provided by Dr. Richard Bucala from Yale University. Rat IgG2b monoclonal antibodies (isotype controls for anti-CD74) were from BD Biosciences (San Jose, CA). AMD3100 and SB225002 were purchased from Tocris (Minneapolis, MN). HE staining reagents were from Fisher Scientific. The rest of the materials used were from Sigma-Aldrich or as described in the methods.

## Statistical analysis

All statistical analyses were performed using R [17]. Differences in 50% threshold between baseline and 24-hr post-treatment were evaluated using paired t-tests (ggpubr package).

Comparisons of 50% threshold scores between groups 24 hr post-treatment (and all group comparisons) were assessed with ANOVA followed by Dunnett's test if the ANOVA was significant. A p < 0.05 was considered statistically significant. Mean and ± SEM are reported.

## Results

### Effect of intravesical pretreatment on PAR4-induced BHA

PBS pretreatment followed by either scrambled peptide (PBS-Scramb; control group with no pain) or PAR4-activating peptide (PBS-PAR4; unalleviated pain group) set up the conditions for testing antagonists on the effect of PAR4-induced BHA.

PBS-Scramb group showed no difference in 50% threshold responses between baseline and 24 hr post-treatment (baseline: 0.214 ± 0.014 g; 24 hr post: 0.203 ± 0.01 g). However, a profound and significant decrease in 50% threshold was observed in the PBS-PAR4 group between baseline and 24 hr and also when compared to PBS-Scramb group (0.005 ± 0.001 g; Fig 2A). This significant difference between scrambled-treated and PAR4-treated groups at 24 hr is consistent with our previous findings that PAR4-AP induced BHA [6, 7].

Pre-treatment with intravesical MIF mAb robustly inhibited PAR4-induced BHA at 24 hr (0.220 ± 0.01 g) while pre-treatment with isotype IgG had no effect on PAR4-induced BHA at 24 hr (0.007 ± 0.003 g; Fig 2B). Similarly, pre-treatment with intravesical CD74 mAb to block CD74 (specific MIF receptor) prevented PAR4-induced BHA at 24 hr (0.183 ± 0.02 g) while pre-treatment with isotype IgG had no effect on PAR4-induced BHA at 24 hr (0.007 ± 0.003 g; Fig 2C). In addition, pre-treatment with MIF098 (to block MIF's interaction with CD74) also partially and significantly decreased PAR4-induced BHA at 24 hr (0.116 ± 0.016 g) while pre-treatment with vehicle had no effect (0.006 ± 0.002 g; Fig 2D). Pre-treatment with intravesical AMD3100 to block CXCR4 receptors was also effective in preventing PAR4-induced BHA at 24 hr (0.216 ± 0.017 g; Fig 2E) while pre-treatment with intravesical SB225002 to block CXCR2 receptors had no effect on PAR4-induced BHA and still showed a profound decrease in VF threshold at 24 hr (0.006 ± 0.002 g; Fig 2F). In summary, pre-treatment with antagonists of MIF and individual MIF receptors CD74 and CXCR4 blocked PAR4-induced BHA while pre-treatment with a CXCR2 antagonist had no effect.

### Intravesical antagonism of MIF receptors block PAR4-induced HMGB1 release in mice

We examined the effect of blocking activation of specific intravesical MIF receptors on PAR4-induced bladder HMGB1 release [16] by measuring levels of HMGB1 in the intravesical fluid after different treatments. In fact, intravesical HMGB1 was significantly increased in PBS-PAR4 treated mice when compared to PBS-Scramb treated mice (Table 1). Pre-treatment with a MIF/CD74 binding inhibitor (MIF098) or with a CXCR4 antagonist (AMD3100) prevented increases in HMGB1 release (Table 1). Pre-treatment with a CXCR2 antagonist (SB225002) on the other hand, was not effective in preventing PAR4 induced HMGB1 release and significant increases were still noted (Table 1).

### Intravesical antagonism of MIF receptors block PAR4-induced HMGB1 and MIF release from human urothelial cells

We also examined the effect of blocking individual MIF receptors on PAR4-induced MIF and HMGB1 release in primary human urothelial cells *in vitro*. Pre-treatment with DMSO (vehicle control) followed by treatment with PAR4 elicited significant release of MIF and HMGB1 from urothelial cells compared to treatment with DMSO followed by scrambled peptpide (as

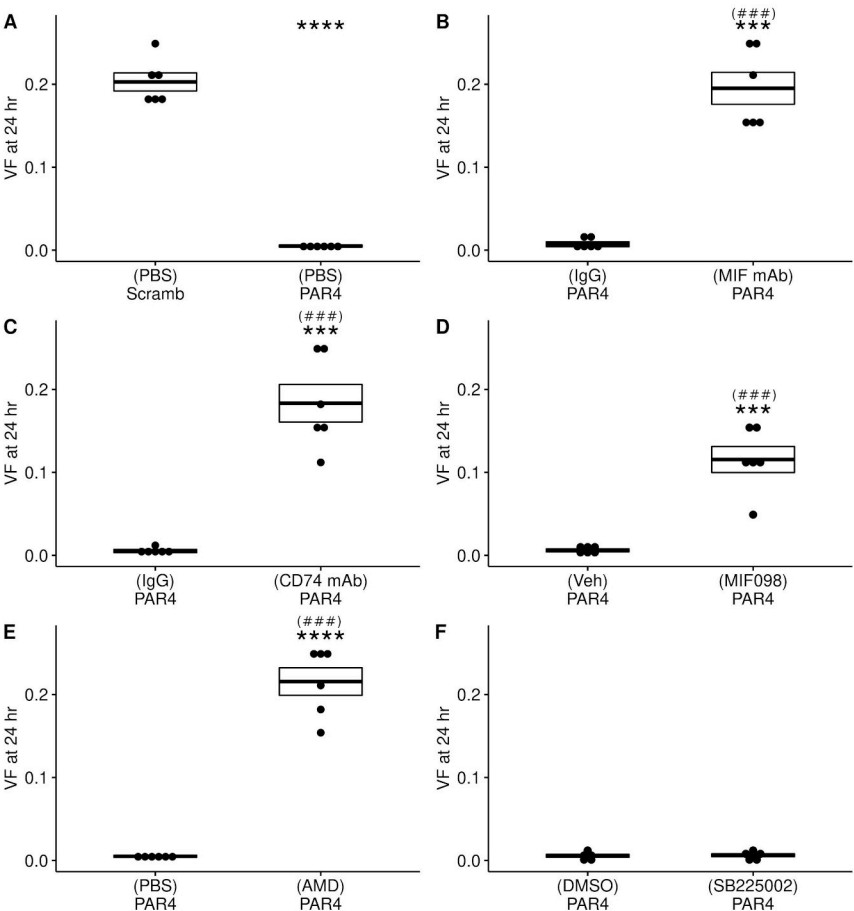

**Fig 2. Intravesical MIF and MIF receptors mediate PAR4-induced bladder hyperalgesia.** Bladder hyperalgesia (BHA; decreases in 50% VF threshold) at 24 hr post treatment were compared across all groups. A) In mice pre-treated with intravesical PBS, intravesical PAR4 peptide resulted in a marked and significant decrease in 50% threshold score when compared to scrambled peptide treatment. B) Pre-treatment with anti-MIF mAb prevented PAR4-induced change in 50% threshold while pre-treatment with isotype mIgG1 had no effect. C) Pre-treatment with anti-CD74 monoclonal antibody also blocked PAR4-induced changes in 50% threshold while pre-treatment with isotype control (rIgG2b) was not effective. D) Pre-treatment with MIF098 (MIF antagonist that prevents binding to CD74) had a significant effect on 50% threshold so that it was still significantly different from the vehicle treated group. E) Pre-treatment with CXCR4 antagonist/AMD3100 followed by PAR4-AP also blocked PAR4-induced changes in 50% threshold. F) Pre-treatment with CXCR2 antagonist/SB225002 followed by PAR4-AP treatment had no effect on 50% threshold and similar results were obtained from mice pre-treated with vehicle, 0.1% DMSO in PBS followed by PAR4 treatment. **** = $p < 0.0001$, *** = $p < 0.001$ when compared to appropriate control group; (###) = $p < 0.001$ when compared to PBS-PAR4 (24 hr).

**Table 1. HMGB1 concentration in intravesical fluid after each treatment.**

| Intravesical Pretreat. | Intravesical Treat. | HMGB1 (pg/ml) |
|---|---|---|
| PBS | *Scrambled peptide* | 147.0 ± 5.1 |
| PBS | *PAR4-AP* | 191.0 ± 12.0* |
| MIF098 | *PAR4-AP* | 177.0 ± 8.5 |
| AMD3100 | *PAR4-AP* | 148.0 ± 14.4 |
| SB225002 | *PAR4-AP* | 186 ± 4.7* |

Mean ± SEM. N = 6 per group.

* = p <0.05 compared to PBS-Scrambled Peptide group.

**Table 2. Concentration of HMGB1 or MIF in culture media following different treatments.**

| Treatment | HMGB1 (pg/ml) | MIF (pg/ml) | Wells |
|---|---|---|---|
| DMSO-Scramb | 753 ± 53.2 | 57 ± 9.1 | 12 |
| DMSO-PAR4 | 1129 ± 80.3 ** | 370 ± 77.5 ** | 18 |
| ISO1-PAR4 | 473 ± 36.6 | 26 ± 6.1 | 6 |
| MIF098-PAR4 | 775 ± 182 | 42 ± 12.9 | 6 |
| PBS-Scramb | 1079 ± 178.0 | 68 ± 14.9 | 6 |
| PBS-PAR4 | 2390 ± 495.0 # | 210 ± 42.5 # | 6 |
| AMD-PAR4 | 1232 ± 133.0 | 223 ± 31.7 ## | 6 |

** = $p < 0.01$ compared to DMSO-Scramb;

# = $p < 0.05$,

## = $p < 0.01$, compared to PBS-Scramb.

control; Table 2). Pre-treatment with a MIF antagonist (ISO-1) or with MIF098 (MIF/CD74 binding inhibitor) prevented both MIF and HMGB1 release from PAR4 stimulation (Table 2). Pre-treatment with a CXCR4 antagonist (AMD3100) blocked PAR4-induced HMGB1 release from primary urothelial cells but not MIF release, since the culture media still showed significantly increased MIF levels when compared to cells pre-treated with PBS followed by a scrambled peptide (as control; Table 2).

## Effect of PAR4 on micturition

**Awake micturition parameters: VSOP.**   We examined the effects of different intravesical pretreatments on micturition parameters in awake mice using the Voided Stain on Paper (VSOP) method as reported [7, 18, 24].

We observed no statistically significant differences between mice pre-treated with PBS and receiving intravesical scrambled peptide or PAR4 in micturition volume (Fig 3; S1 Table) or frequency (Fig 4; S1 Table), which is consistent with our previous studies without PBS pre-treatment [7]. None of the other treatments had a significant effect on awake micturition volume or frequency (Figs 3 and 4; S1 Table).

**Anesthetized micturition parameters: CMG.**   We also examined the effect of our model of BHA on bladder function using cystometry in anesthetized mice. Single cystometrograms (CMG) showed similar profiles between PBS-Scramble and PBS-PAR4-AP treatment groups (Fig 5A and 5B). In fact, Table 3 shows that similar values with no significant differences between the two groups for all parameters recorded during single CMG. Similarly, continuous CMG showed similar profiles between the two groups (Fig 5C and 5D) and the intercontraction interval measured from the continuous CMG was not significantly different between control group and PAR4-AP treatment group (n = 8) with 279 ± 35.1 sec vs 336 ± 61.9 sec, $p > 0.05$ (Table 3).

## Effect of PAR4 and intravesical pre-treatments on histological bladder changes

H&E stained bladder sections from all groups were examined by a pathologist blinded to the treatment and scored for edema and inflammation changes. (Fig 6). Intravesical treatment with PAR4 did not result in an increase in edema or inflammation when compared to treatment with scrambled peptide in PBS-treated animals (Table 4). This is consistent with our

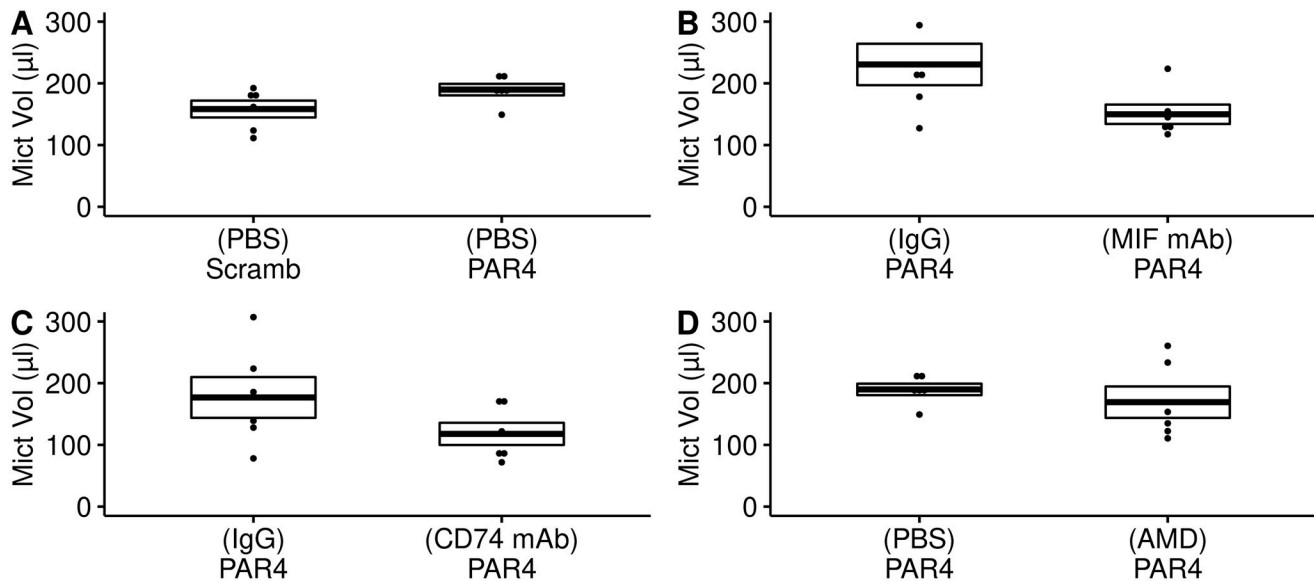

**Fig 3. Awake micturition volume in treatment groups.** Micturition volume ($\mu$l) observed using awake Voided Stain on Paper (VSOP) for all treatment groups and analyzed using ANOVA followed by post-hoc Dunnett tests. There were no significant differences between the groups when compared to control (no pain; PBS-pretreatment, scrambled peptide-treatment; PBS-Scramb). A) Micturition volume in mice pre-treated with intravesical PBS followed by intravesical scrambled PAR4 peptide or PAR4-AP. B) Pre-treatment with isotype mIgG1 (IgG) antibody or anti-MIF mAb or anti-MIF mAb, both groups treated with PAR4 to elicit bladder pain. C) Pre-treatment with isotype control (rIgG2b; IgG) or anti-CD74 monoclonal antibody and then treated with intravesical PAR4. D) Pre-treatment with PBS or CXCR4 antagonist AMD3100 (AMD) followed by PAR4-AP.

previous reports [6, 8, 16, 19]. Moreover, none of the other intravesical treatments resulted in significant changes in edema when compared to scrambled peptide treatment (Table 4). Treatment with isotype IgG caused a statistically significant mild increase in inflammation when compared to PBS-Scrambled treatment (Table 4).

## Effect of PAR4 on inflammatory cytokines in the bladder

We assessed whether PAR4 induced bladder pain is accompanied by activation of inflammatory pathways in the bladder by investigating changes in mRNA expression and protein levels of two pro-inflammatory cytokines IL-1$\beta$ and TNF-$\alpha$.

We normalized pro-inflammatory cytokine mRNA expression in the bladder to the expression of the three most stable and robust housekeeping genes (heat shock protein, 18S and actin) out of a panel of five housekeeping genes tested to increase the reproducibility of the results [25]. We observed no changes in mice treated with PBS-Scramb (no pain control) and those treated with PBS-PAR4 (unalleviated pain group). The $\Delta$CT for IL-1$\beta$ between PBS-Scramb and PBS-PAR4 groups was not significantly different (14.9 $\pm$ 0.47 vs 14.7 $\pm$ 0.45, p>0.05; Panel A in S1 Fig; fold-change = 1.1). Similarly, no differences in expression of TNF-$\alpha$ in the bladder was noted between the two groups ($\Delta$CT = 15.5 $\pm$ 0.67 vs. 15.4 $\pm$ 0.96, p>0.05; PBS-Scramb vs. PBS-PAR4, respectively; Panel B in S1 Fig; fold-change = 0.9).

Normalized to total amount of protein, the IL-1$\beta$ levels in the bladder of PBS-PAR4 treatment group was 10.1 $\pm$ 0.8 pg/mg while the PBS-Scramb group was 18.6 $\pm$ 2.1 pg/mg (p < 0.05) showing a significant decrease (Panel A in S2 Fig). Likewise, the level of TNF-$\alpha$ in PAR4-AP treatment group was significantly decreased when compared to the PBS-Scramb group (4.98 $\pm$ 1.90 vs 13.9 $\pm$ 2.10 pg/mg; p < 0.05; Panel B in S2 Fig).

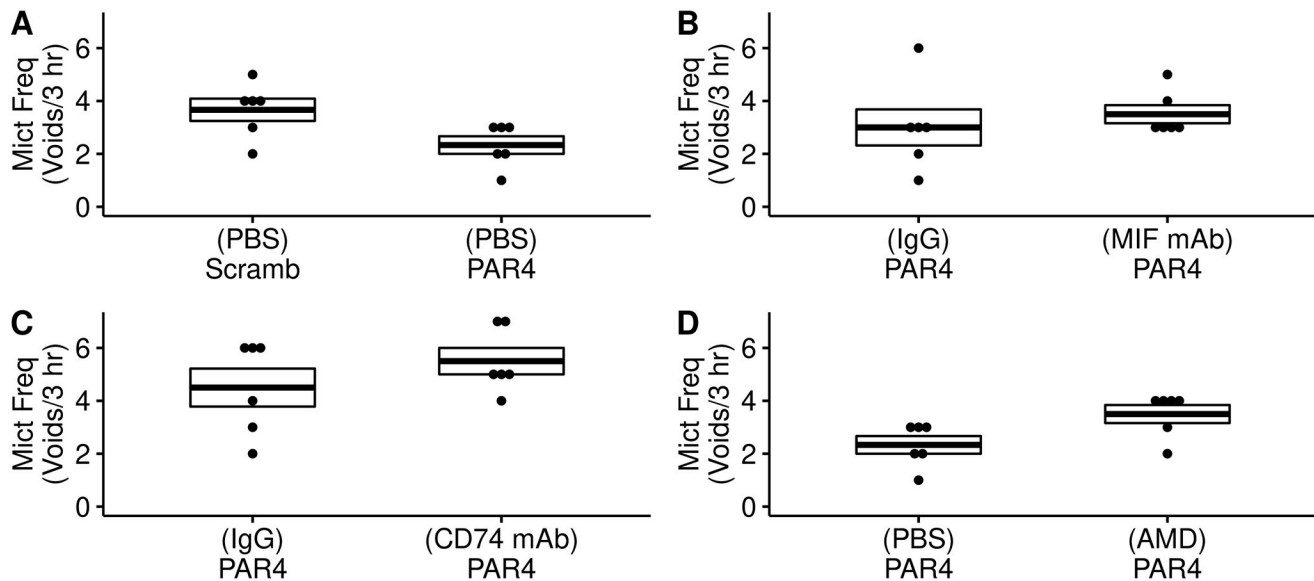

**Fig 4. Awake micturition frequency in treatment groups.** Micturition frequency (number of voids in 3 hour observation period) recorded using awake Voided Stain on Paper (VSOP) for all treatment groups and analyzed using ANOVA followed by post-hoc Dunnett tests. There were no significant differences between the groups when compared to control (no pain; PBS-Scramb). A) Micturition frequency in mice pre-treated with intravesical PBS followed by intravesical scrambled PAR4 peptide or PAR4-AP. B) Pre-treatment with isotype mIgG1 antibody (IgG) or anti-MIF mAb, both groups treated with PAR4 to elicit bladder pain. C) Pre-treatment with isotype control (rIgG2b; IgG) or anti-CD74 monoclonal antibody and then treated with intravesical PAR4. D) Pre-treatment with PBS or CXCR4 antagonist AMD3100 (AMD) followed by PAR4-AP.

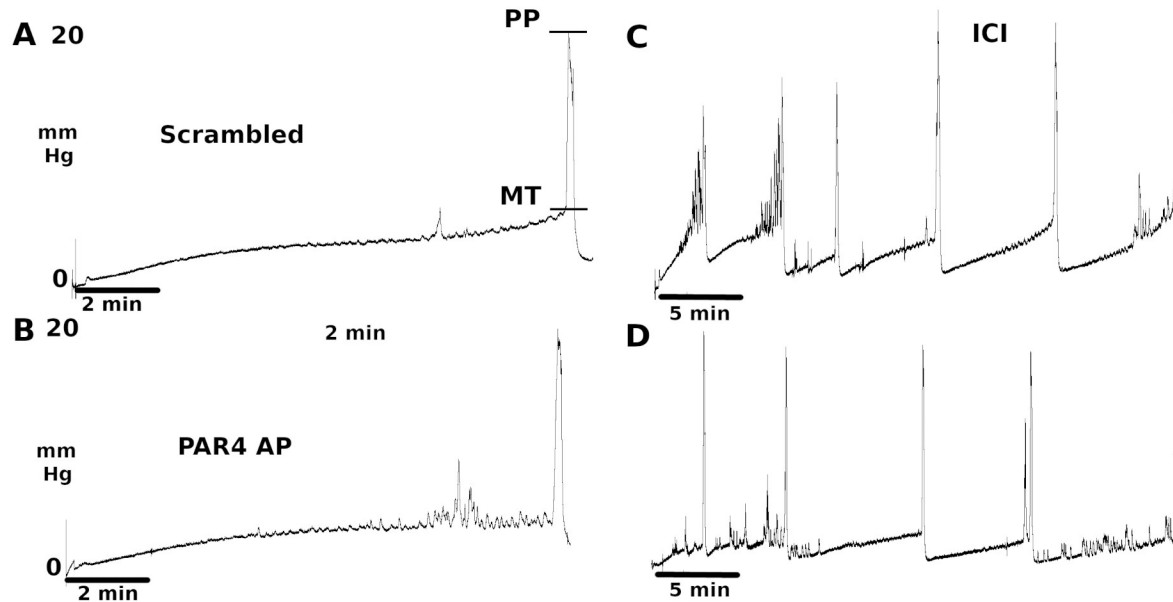

**Fig 5. Anesthetized cystometry showed no effect from PAR4 treatment.** Mice were instrumented with a transvesical bladder catheter as described that allowed for filling of the bladder and recording of bladder pressure. Single cystometrograms (A; B) showed very similar profiles between PBS-Scrambled treated and PBS-PAR4-AP treated mice. Continuous cystometry (C; D) also showed no difference in profiles between the two groups. MT = micturition threshold; PP = peak pressure; ICI = intercontraction interval.

**Table 3. Anesthetized micturition parameters mice pre-treated with PBS followed by scrambled peptide or PAR4 Activating peptide (AP).**

| Intravesical Pretreat. | *PBS* | *PBS* |
|---|---|---|
| Intravesical Treat. | *Scrambled peptide* | *PAR4-AP* |
| **Single CMG (N = 8)** | | |
| Micturition Threshold (mm Hg) | 8 ± 0.9 | 8 ± 0.6 |
| Peak Pressure (mm Hg) | 21 ± 0.8 | 20 ± 0.8 |
| Infused Volume ($\mu$l) | 84 ± 14.3 | 107 ± 23.2 |
| Micturition Volume ($\mu$l) | 57 ± 11 | 73 ± 18.0 |
| Efficiency (%) | 67 ± 4.7 | 68 ± 4.0 |
| **Continuous CMG (N = 8)** | | |
| Intercontraction Interval (sec) | 279 ± 35.1 | 336 ± 61.9 |

## Discussion

Our current results clearly show that MIF mediates PAR4-induced bladder pain through interaction with MIF receptors. The effects of PAR4 on pain are complex. Activation of PAR4 receptors is associated with inhibition of hyperalgesia in some systems, for example, bowel [27] and promote hyperalgesia in knee [28, 29], like we see in bladder. Thus it appears that the effects of PAR4 on sensation are likely tissue/organ and dose dependent [30]. Therefore, it is important to investigate the mechanisms of PAR4 mediated pain in the bladder, specifically, as it may shed some light on possible targets for bladder pain states.

Our current results confirm earlier findings (using systemic MIF antagonists [6] or MIF-deficient mice [20]) and show that MIF mediates PAR4-induced BHA in mice. In addition, we extend our previous findings by localizing MIF's site of action to the bladder since intravesical MIF mAb (but not control isotype) completely blocked PAR4-induced BHA. Therefore, we establish that MIF, working at the level of the bladder, mediates bladder pain in this model.

Moreover, we provide evidence that specific urothelial MIF receptors mediate PAR4-induced BHA. Antagonism of CD74, the cognate MIF receptor, with intravesical anti-CD74 mAb completely prevented while MIF098 partially blocked PAR4-induced BHA. The difference between the efficacy of these two CD74 antagonists may reflect dosing differences between the agents or the intrinsic avidity of the bivalent antibody. Alternatively, there may be a difference blocking effectiveness since mAb will block CD74 receptor at urothelium while MIF098 will block the CD74 binding site on MIF [31] in the intravesical fluid.

Similarly, AMD3100, an intravesical antagonist to CXCR4 (MIF co-receptor [32] that also binds to CXCL12 [10]) also blocked PAR4-induced BHA. This last observation confirms and extends previous findings using systemic administration of AMD3100 [6] and clearly localizes the effect to the bladder. Intravesical antagonism of CXCR2 receptors (MIF co-receptor [32] that also binds to CXCL8) was not effective in preventing PAR4-induced BHA suggesting that CXCR2 receptors are not involved in mediating bladder pain in this model.

We previously showed that activation of PAR4 receptors elicited release of urothelial HMGB1 in mice and in transformed benign human urothelial cells [6]. Our current results confirm these findings *in vivo* and *in vitro* (in human primary urothelial cells). In addition, our current results show that MIF mediates PAR4-induced HMGB1 release since it was blocked by an intravesical CD74 coupling inhibitor (MIF098) or an intravesical CXCR4 antagonist (AMD3100) while an intravesical CXCR2 antagonist had no effect. In human urothelial cells, MIF mediates PAR4-induced HMGB1 release through activation of CD74 receptors

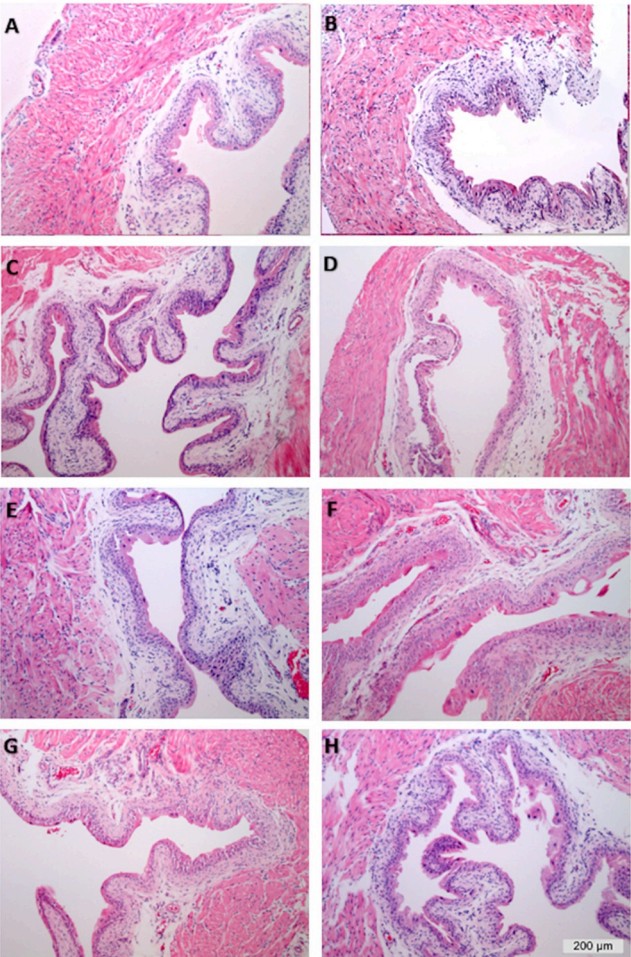

**Fig 6. PAR4 treatment does not induce significant changes in bladder edema or inflammation.** Representative sections of H&E bladder sections from all groups. A) PBS-Scrambled peptide treatment caused mild edema likely due to catheter insertion. B) No significant difference in edema or inflammation seen in PBS-PAR4 treatment. Pre-treatment with isotype IgG (mIgG1; C or rIgG2b; E) caused a small but statistically significant increase in inflammation when compared to PBS-Scrambled group. No significant changes in edema or inflammation were noted for pre-treatment with anti-MIF mAb (D), anti-CD74 mAb (F), vehicle for MIF098 (G), MIF098 (H), AMD3100 (CXCR4 antagonist; I) or SB225002 (CXCR2 antagonist: J). Calibration bar = 200 $\mu$m.

only. Lastly, we confirm in primary human urothelial cells that MIF also works through CD74 to trigger MIF release in an autocrine manner, as we reported in a rodent model [12]. In this manner, MIF can maintain pain in a feedback mechanism that regulates MIF and HMGB1 release.

PAR4 is located in urothelial cells in the bladder and also in nerves in the bladder [1, 33]. In fact, the same dose of intravesical PAR4 used in our current study activated bladder afferents in mice causing a significant and sustained increase in afferent firing [34]. Thus intravesical PAR4 may stimulate bladder afferent nerves directly to result in hyperalgesia.

Our current results suggest that PAR4 may also have an indirect effect that promotes BHA. In this case, intravesical PAR4 activates urothelial PAR4 receptors to release urothelial MIF. Released MIF then activates urothelial MIF receptors (CD74 and CXCR4 in mice; CD74 only in humans) to trigger release of urothelial HMGB1. Released HMGB1 then activates urothelial

**Table 4. Bladder histological changes (N = 6) in all experimental groups.**

| Intravesical Pretreat. | Intravesical Treat. | Edema | Inflammation |
|---|---|---|---|
| *Acute BHA model* | | | |
| PBS | *Scrambled peptide* | 1.0 ± 0.0 | 0.1 ± 0.1 |
| PBS | *PAR4-AP* | 1.0 ± 0.0 | 0.8 ± 0.2 |
| *MIF antagonism* | | | |
| mIgG1 | *PAR4-AP* | 0.8 ± 0.2 | 1.3 ± 0.3* |
| Anti-MIF mAb | *PAR4-AP* | 0.9 ± 0.3 | 0.7 ± 0.2 |
| *CD74 antagonism* | | | |
| rIgG2b | *PAR4-AP* | 1.3 ± 0.4 | 1.5 ± 0.4* |
| Anti-CD74 mAb | *PAR4-AP* | 1.3 ± 0.3 | 1.0 ± 0.3 |
| Vehicle | *PAR4-AP* | 0.8 ± 0.4 | 0.3 ± 0.3 |
| MIF098 | *PAR4-AP* | 1.3 ± 0.2 | 0.3 ± 0.2 |
| *CXCR4 antagonism* | | | |
| AMD3100 | *PAR4-AP* | 1.2 ± 0.2 | 0.9 ± 0.4 |
| *CXCR2 antagonism* | | | |
| 0.1% DMSO | *Scrambled peptide* | 1.8 ± 0.4 | 0.8 ± 0.3 |
| SB225002 | *PAR4-AP* | 1.7 ± 0.3 | 1.1 ± 0.3 |

TLR4 receptors to result in bladder hyperalgesia [8]. This indirect effect lasts longer (24 hr) than the direct effect of PAR4 on bladder nerves and can be extended to several days (up to 9 days) with repeated application of intravesical PAR4 activation [18, 19].

None of the treatments resulted in a significant difference in edema when compared to intravesical scrambled peptide (control; no pain group). There was a mild and significant increase in inflammation in mice treated with isotype IgG only, perhaps indicating a mild allergic reaction. Once again, our results confirm that PAR4 induces BHA with minimal or no changes in histology to indicate bladder inflammation [6, 7, 16], an observation confirmed by a separate group [35].

Similarly, none of the treatments resulted in a significant difference in micturition volume or frequency in awake mice when compared to mice treated with intravesical scrambled peptide in agreement with our earlier reports [6, 7, 16]. Anesthetized cystometry also failed to detect differences between animals treated with PAR4 and those with scrambled peptide. Therefore, we provide additional evidence that intravesical PAR4 elicits bladder pain without changing either awake or anesthetized micturition parameters.

We further examined inflammatory changes in the bladder by measuring changes in expression and protein levels of two pro-inflammatory cytokines (IL-1$\beta$ and TNF-$\alpha$). These inflammatory markers are well-recognized markers of bladder inflammmation in rodents since both protein levels and mRNA levels increase in the bladder in different models of bladder inflammation (including chemical cystitis, e.g. cyclophosphamide/ifosfamide; common methods of experimental cystitis) [36–41]. Thus measuring protein and mRNA levels in the bladder is a reliable and accepted method for examing inflammatory changes in the bladder caused by our bladder pain model. We found no evidence that either protein or mRNA levels of IL-1$\beta$ or TNF-$\alpha$ increased after intravesical PAR4. Moreover, protein levels of both cytokines were decreased in bladder tissue 24 hr after PAR4 treatment. It should be noted that down regulation of IL1-$\beta$ and TNF-$\alpha$ after activation of PAR4 has been reported [42, 43].

Taken together, our current findings failed to show evidence that PAR4-induced BHA also results in micturition changes (awake or anesthetized cystometry), bladder inflammation, as

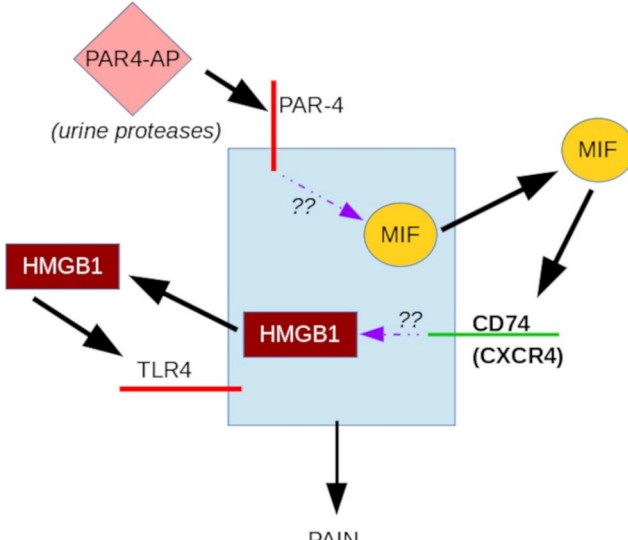

**Fig 7. Diagram of possible mechanism for MIF-CD74-HMGB1 mediated bladder pain.** PAR4-activating peptide (PAR4-AP), and more likely, urine proteases activate PAR4 receptors found in the urothelium. This event triggers a release of urothelial MIF to the extracellular (and even intravesical) space where MIF can bind CD74 and/or CXCR4 (MIF receptors) to mediate urothelial HMGB1 release. Once released, HMGB1 can bind HMGB1 receptors (most likely intravesical TLR4) to mediate bladder pain. Exact, detailed mechanisms are not known as this point and indicated by question marks.

determined by changes in common bladder pro-inflammatory cytokines or histological evidence of edema/inflammation in the bladder. Thus, it appears that intravesical PAR4 elicits bladder pain without significant bladder damage that would result in inflammatory changes in the bladder.

In summary, our current results show that PAR4-induced bladder pain may be modulated by intravesical antagonists of MIF and of specific MIF receptors, namely CD74 (cognate MIF receptor) and CXCR4 (Fig 7). We confirmed that MIF is directly involved in the release of urothelial HMGB1 likely mediating bladder pain through intravesical TLR4 receptors as we demonstrated earlier [8]. It remains to be determined whether the mechanism for bladder pain outlined here is also at work in other types of bladder pain. It should be noted that elevated urine levels of MIF were observed in patients with radiation cystitis, urinary tract infections and interstitial cystitis with Hunner lesions [44]. Thus, future investigations are needed to examine mechanisms of bladder pain in other conditions and to elucidate possible therapeutic avenues in painful bladder conditions.

## Conclusion

Activation of intravesical PAR4 receptors elicits urothelial MIF release that then binds to urothelial MIF receptors to promote release of urothelial HMGB1 to result in bladder pain by activation of intravesical HMGB1 receptors (most likely TLR4). This mechanism of bladder pain warrants further investigation as it may occur in other causes of bladder pain.

## Supporting information

**S1 Table. Awake micturition parameters.**
(PDF)

**S1 Fig. Bladder pPCR: TNF-$\alpha$, IL-1$\beta$.**
(TIF)

**S2 Fig. Bladder protein: TNF-$\alpha$, IL-1$\beta$.**
(TIF)

**S1 Dataset. Original datasets.**
(XLSX)

## Author Contributions

**Conceptualization:** Fei Ma, Katherine L. Meyer-Siegler, Pedro L. Vera.

**Data curation:** Shaojing Ye.

**Formal analysis:** Shaojing Ye, Fei Ma, Dlovan F. D. Mahmood, Katherine L. Meyer-Siegler, Pedro L. Vera.

**Funding acquisition:** Richard Bucala, Pedro L. Vera.

**Investigation:** Shaojing Ye, Fei Ma, Dlovan F. D. Mahmood, Katherine L. Meyer-Siegler, Raymond E. Menard, David E. Hunt.

**Methodology:** Shaojing Ye, Fei Ma, Pedro L. Vera.

**Project administration:** Pedro L. Vera.

**Resources:** Lin Leng, Richard Bucala.

**Supervision:** Pedro L. Vera.

**Validation:** Pedro L. Vera.

**Writing – original draft:** Shaojing Ye, Fei Ma, Richard Bucala, Pedro L. Vera.

**Writing – review & editing:** Shaojing Ye, Fei Ma, Dlovan F. D. Mahmood, Katherine L. Meyer-Siegler, Raymond E. Menard, David E. Hunt, Lin Leng, Richard Bucala, Pedro L. Vera.

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
