## [Decision Letter · Decision Letter 0]

23 Jun 2021

PONE-D-21-10916

Intravesical macrophage migration inhibitory factor (MIF) receptors mediate bladder pain

PLOS ONE

Dear Dr. Vera,

Thank you for submitting your manuscript to PLOS ONE. After careful consideration, we feel that it has merit but does not fully meet PLOS ONE’s publication criteria as it currently stands. Therefore, we invite you to submit a revised version of the manuscript that addresses the points raised during the review process.

ACADEMIC EDITOR:

Specific point 3.) (Ibudilast) as mentioned by Reviewer #1 under 5.) Review Comments to the Author may be either addressed by experiments or expanded discussion.

We look forward to receiving your revised manuscript.

Kind regards,

Jürgen Bernhagen

Academic Editor

PLOS ONE

2. As part of PLOS ONE's publication criteria, the journal requires that in each submission, experiments, statistics, and other analyses are performed to a high technical standard and are described in sufficient detail (https://journals.plos.org/plosone/s/criteria-for-publication).  To this end, please revise your Methods section to address the following: (1) the number of animals in each group and how you determined the sample size; (2) a complete description of methods undertaken to ameliorate pain and distress (e.g. anesthetics and analgesics, supportive care, etc.); (3) details of monitoring frequency and parameters; (4) details about humane endpoints for any animals who became severely ill during the study; (5) unanticipated adverse events and how they were managed; (6) the rate of mortality during the study and the cause of death (if applicable).

“This study is funded by NIH (DK121695, PLV; AR049610, RB). The material is the 366 result of work supported with the resources and facilities at the Lexington (Kentucky) 367 VA Health Care System.”

 “This study is funded by NIH (DK121695, PLV; AR049610, RB). The material is the 366 result of work supported with the resources and facilities at the Lexington (Kentucky) 367 VA Health Care System.”

Additional Editor Comments (if provided):

Comments Academic Editor

In addition to the comments provided by the referees, please consider creating a summarizing scheme/cartoon that sums up the "induction - vesicle - MIF - receptor - HMGB1 pathway". This would greatly help readers from related fields.

Reviewers' comments:

Reviewer's Responses to Questions

**Comments to the Author**

1. Is the manuscript technically sound, and do the data support the conclusions?

Reviewer #1: Yes

2. Has the statistical analysis been performed appropriately and rigorously? 

Reviewer #1: Yes

3. Have the authors made all data underlying the findings in their manuscript fully available?

Reviewer #1: Yes

4. Is the manuscript presented in an intelligible fashion and written in standard English?

Reviewer #1: Yes

5. Review Comments to the Author

Reviewer #1: In this manuscript (PONE-D-21-10916) Ye, Ma, et al. utilize a series of pharmacological reagents (mainly inhibitors) to demonstrate the role of MIF signaling and its associated receptors in mouse models of bladder pain. This is a great follow up to the work of the Vera laboratory and now reveals a role for CD74 in MIF-mediated bladder pain, as well as confirming previous results for CXCR4. The manuscript is generally well presented and sets up the next set of translational studies well. With that said, there are some opportunity to improve the clarity of the message and robustness of the data presented.

1. The title is a bit vague and does not capture the new role for CD74. I would recommend being clear about this as it will get readers attention easier.

2. The abstract also should be revised to include this key finding, as well as the in vivo functional observations (which are not clearly articulated).

3. Given the use of ibudilast (which blocks among other interactions, CD74/MIF) in multiple clinical settings, it would be very useful to use it in these studies. While it may be out of the scope of these studies, this may also be a good use of the CD74 KO mice (developed by one of the co-authors, Dr. Bucala).

4. The data in Table 3 are useful, but would be worth presented as a series of graphs. The authors should use their discretion as to how many conditions they show as the rest can be included as supplemental data (in the form of a table).

5. The immune changes presented are generally qualitative in terms of histology. It would be useful to quantify these, as well as other markers of inflammation via IHC. While it may end up being a negative result (based on the previous Vera laboratory report (PMID 31289792), it would still be worth assessment with appropriate quantification.

6. The qPCR data on IL1-b and TNF-a, while negative, should still be graphed in some manner (as a supplemental figure is sufficient).

7. In the conclusions, the authors should be more precise about “TLR4” as it is suggested that this is only receptor for HMGB1 (which can bind to other receptors, including TLR2 and RAGE). It may be worth just switching “TLR4” with “HMGB1” receptors or ending the sentence as such: …HMGB1 to result in bladder pain.”

6. PLOS authors have the option to publish the peer review history of their article (what does this mean?). If published, this will include your full peer review and any attached files.

---

## [Author Response · Author response to Decision Letter 0]

16 Jul 2021

We thank the reviewer of our manuscript for the helpful and insightful comments. We addressed the comments as indicated below and consider the manuscript improved. Changes are indicated in the text in the following manner: A) deleted text: crossed out in red: B) New text: in blue with wavy underline.

Changes made in response to Reviewer 1:

1. The title is a bit vague and does not capture the new role for CD74. I would recommend being clear about this as it will get readers attention easier.

a. Title was amended to clearly name the MIF receptors

2. The abstract also should be revised to include this key finding, as well as the in vivo functional observations (which are not clearly articulated).

a. Abstract was revised to highlight the role of CD74 and the functional observations have been made clearer.

3. Given the use of ibudilast (which blocks among other interactions, CD74/MIF) in multiple clinical settings, it would be very useful to use it in these studies. While it may be out of the scope of these studies, this may also be a good use of the CD74 KO mice (developed by one of the co-authors, Dr. Bucala).

a. This is an excellent suggestion and we plan to follow up on this in future studies. As this likely represents a separate study we will continue with our current findings having blocked CD74 interactions in two ways: with anti-CD74 mAb and MIF98.

4. The data in Table 3 are useful, but would be worth presented as a series of graphs. The authors should use their discretion as to how many conditions they show as the rest can be included as supplemental data (in the form of a table).

a. Two additional graphs are included:

i. Micturition volume (now Fig 3)

ii. Micturition frequency (now Fig 4)

b. Full table, with all treatment groups is now a supplemental table (Suppl. Table 1)

5. The immune changes presented are generally qualitative in terms of histology. It would be useful to quantify these, as well as other markers of inflammation via IHC. While it may end up being a negative result (based on the previous Vera laboratory report (PMID 31289792), it would still be worth assessment with appropriate quantification.

a. This is a very interesting suggestion and worth following up in future studies. We highlight the quantitative (ELISA) assessments of bladder protein levels of TNF-alpha and IL-1B by adding a supplemental figure (see 5b below). We also extended our discussion and added references to show these are well accepted markers of bladder inflammation (as reported in the literature for other bladder inflammatory models) and we do not see a change in bladder levels of these markers.

b. A figure (Supplemental Fig 2) is presented as supplemental evidence, showing bladder levels of these two inflammatory markers as determined by ELISA.

6. The qPCR data on IL1-b and TNF-a, while negative, should still be graphed in some manner (as a supplemental figure is sufficient).

a. Excellent suggestion. Added as a supplemental figure (Supl. Fig 1).

7. In the conclusions, the authors should be more precise about “TLR4” as it is suggested that this is only receptor for HMGB1 (which can bind to other receptors, including TLR2 and RAGE). It may be worth just switching “TLR4” with “HMGB1” receptors or ending the sentence as such: …HMGB1 to result in bladder pain.”

a. Concluding statement states “HMGB1 receptors”. We have retained the notion that most likely this is mediated by intravesical TLR4 receptors based on our experimental evidence as cited in the text.

Changes made in Response to Academic Editor.

1. …please consider creating a summarizing scheme/cartoon that sums up the "induction - vesicle - MIF - receptor - HMGB1 pathway". This would greatly help readers from related fields.

a. Thank you for this suggestion. A figure has been added as part of the summary paragraph. (Fig. 7)

In addition, while animal numbers had been listed through the text and figure captions, they are now also noted in Methods, as requested.

Finally, Funding information was removed from the Acknowledgements section as requested. Funding statement should read:

“This study is funded by NIH (DK121695, PLV; AR049610, RB). The material is the result of work supported with the resources and facilities at the Lexington (Kentucky) VA Health Care System.”

---

## [Editor Report · Decision Letter 1]

28 Jul 2021

Intravesical CD74 and CXCR4, macrophage migration

inhibitory factor (MIF) receptors, mediate bladder pain

PONE-D-21-10916R1

Dear Dr. Vera,

We’re pleased to inform you that your manuscript has been judged scientifically suitable for publication and will be formally accepted for publication once it meets all outstanding technical requirements.

Kind regards,

Jürgen Bernhagen

Academic Editor

PLOS ONE

Additional Editor Comments (optional):

Authors have addressed the concerns raised.

Reviewers' comments:

N/A

---

## [Editor Report · Acceptance letter]

13 Aug 2021

PONE-D-21-10916R1 

Intravesical CD74 and CXCR4, macrophage migration inhibitory factor (MIF) receptors, mediate bladder pain 

Dear Dr. Vera:

I'm pleased to inform you that your manuscript has been deemed suitable for publication in PLOS ONE. Congratulations! Your manuscript is now with our production department. 

Kind regards, 

on behalf of

Jürgen Bernhagen 

Academic Editor

PLOS ONE